# Clustering Effect of
# (Linearized) Adversarial Robust Models

**Yang Bai**[1]* **Xin Yan**[2]* **Yong Jiang**[1,2,5] **Shu-Tao Xia**[2,5]† **Yisen Wang**[3,4]†
[1]Tsinghua Berkeley Shenzhen Institute, Tsinghua University
[2]Tsinghua Shenzhen International Graduate School, Tsinghua University
[3]Key Lab. of Machine Perception, School of Artificial Intelligence, Peking University
[4]Institute for Artificial Intelligence, Peking University
[5]PCL Research Center of Networks and Communications, Peng Cheng Laboratory, China

## Abstract

Adversarial robustness has received increasing attention along with the study of adversarial examples. So far, existing works show that robust models not only obtain robustness against various adversarial attacks but also boost the performance in some downstream tasks. However, the underlying mechanism of adversarial robustness is still not clear. In this paper, we interpret adversarial robustness from the perspective of linear components, and find that there exist some statistical properties for comprehensively robust models. Specifically, robust models show obvious hierarchical clustering effect on their linearized sub-networks, when removing or replacing all non-linear components (*e.g.*, batch normalization, maximum pooling, or activation layers). Based on these observations, we propose a novel understanding of adversarial robustness and apply it on more tasks including domain adaption and robustness boosting. Experimental evaluations demonstrate the rationality and superiority of our proposed clustering strategy. Our code is available at https://github.com/bymavis/Adv_Weight_NeurIPS2021.

## 1 Introduction

Nowadays, deep neural networks (DNNs) have shown a strong learning capacity through a huge number of parameters and diverse structures [16, 31, 11]. Meanwhile, adversary has raised increasing security concerns on DNNs due to the observation of adversarial examples (*i.e.,* the examples mislead a classifier when crafted with human imperceptible but carefully designed perturbations) [13]. Adversarial robustness and defense techniques have thus become crucial for deep learning, yet many adversarial defense techniques are found with deficiencies such as obfuscated gradient [2]. Up to now, the widely accepted techniques to improve adversarial robustness are adversarial training variants [23, 32, 38, 33, 35, 36], by training a DNN after data augmentation with the worst-case adversarial examples. Recent study has found that not only do robust models show moderate robustness under vast attacks, but also they can surprisingly boost the downstream tasks (*e.g.*, domain adaption task with subpopulation shift) [29].

However, the underlying mechanism of adversarial robustness is still not clear. Existing studies have explored adversary on some specific components of DNNs, such as batch normalization [12], skip connection [34], or activation layers [14, 4], which yield some enlightening understanding. On the other hand, despite variation in DNNs' structure, the components can be roughly divided into linear and non-linear ones. The non-linear components tend to be instance-wise. For example, adversarial

---

*Equal contribution.
†Correspondence to: Yisen Wang (yisen.wang@pku.edu.cn) and Shu-Tao Xia (xiast@sz.tsinghua.edu.cn).

35th Conference on Neural Information Processing Systems (NeurIPS 2021) .

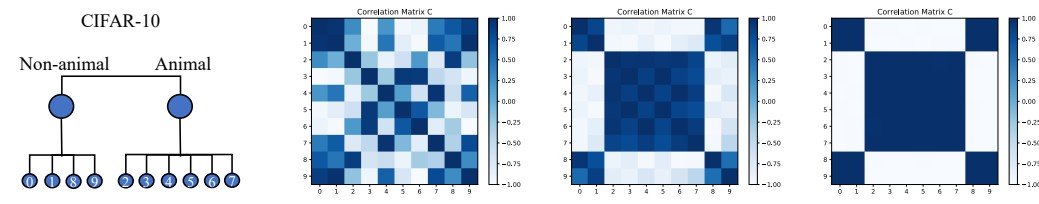

|     |     |     |     |
|-----|-----|-----|-----|
| (a) Class Hierarchy | (b) VGG16-STD | (c) VGG16-AT | (d) VGG16-AT+C |

Figure 1: Correlation matrices $C$ of VGG-16 on CIFAR-10: b) 'STD' and c) 'AT' indicate standard and adversarial training, d) 'AT+C' indicates applying hierarchical clustering in adversarial training. Robust models tend to show clustering effect aligned with a) the class hierarchy of CIFAR-10, which is enhanced in d) our clustering adversarially trained model.

and benign examples tend to present different patterns on activation [4]. In contrast, for those linear components, once the optimization is done and model evaluation mode is activated, they are shared and fixed independently from inputs. Nowadays, the intrinsic relationship of adversary with linearity has been studied. Some ascribes the adversary to linear accumulation of perturbations from inputs to final outputs, dubbed 'accidental steganography' [13]. Some show that linear backward propagation enhances adversarial transferability [34, 14]. Nevertheless, such mentioned linearity analyses are all on the original non-linear models, which hinders a deeper understanding on linearity directly. Further essential exploration is encouraged.

In this paper, we thus study adversarial robustness on the statistical regularity of linear components. We provide a novel insight on adversarial robustness by showing the clustering effect well aligned with *class hierarchy*. As shown in Fig. 1(d), given a data set, superclasses are defined by grouping semantically similar subclasses together to form a hierarchy, *e.g.*, in CIFAR-10, 'Cat' and 'Dog' represent different subclasses/fine labels, despite both being the same superclasses/coarse labels 'Animal'. Based on this, we conduct backward propagation on linearized sub-networks of robust and non-robust DNNs (through adversarial/standard training respectively) to extract the corresponding implicit linear expression. In this fashion, we could obtain a $D_{\text{input}} \times D_{\text{output}}$ linear matrix $W$, which is extracted from inputs to predictive outputs. $D_{\text{input}}$ represents for the dimension of one input data, specifically defined as width $\times$ height $\times$ channel. $D_{\text{output}}$ represents for the dimension of one output vector, which is also the number of classes. We further study the correlation of vectors in $W$ by matrix $C$, whose shape is $D_{\text{output}} \times D_{\text{output}}$. The details of $W$ and $C$ are introduced in Sec. 3.2. Then we find that robust models tend to show hierarchical clustering effect in correlation matrix $C$, which is surprisingly consistent with class hierarchy. For example, in Fig. 1 on CIFAR-10, the fine labels (0,1,8,9) belong to 'Non-animal', others belong to 'Animal'. The matrix $C$ of robust model shows block clustering: values are close to '+1' in the two superclasses (0,1,8,9) and (2,3,4,5,6,7), and values are close to '-1' across these two superclasses. Such clustering effect is enhanced by our clustering regularization penalty in Fig. 1(d). This phenomenon could be semantically explained as subclasses from the same superclass share more feature similarities. Such hierarchical clustering effect can give a novel and insightful explanation on the superiority of robust models in some observed downstream tasks, as they can extract more semantic and representative features. To further confirm our findings, we enhance the hierarchical clustering effect in both adversarial robustness and downstream tasks, *e.g.*, domain adaption proposed in [29, 28]. The improvements of experimental results demonstrate that the superiority of robust models is closely related to their hierarchical classification capability. Our contributions are summarized as follows:

- To the best of our knowledge, we are the first to systematically analyze the statistical regularity of adversarially robust models (through adversarial training) compared to non-robust models (through standard training) on their linearized sub-networks.

- We present an intriguing phenomenon of hierarchical clustering effect in robust models, and provide a novel yet insightful understanding of adversarial robustness. The clustering effect aligned with class hierarchy demonstrates more semantic and representative feature extraction capacity of robust models, which benefits a lot in various tasks.

- Based on the observations, we propose a plugged-in hierarchical clustering training strategy to generally enhance adversarial robustness and investigate some intriguing adversarial attack findings. Besides adversarial-related study, we further explore some downstream tasks with

the understanding of hierarchical clustering, *e.g.*, domain adaption with subpopulation shift. Experimental results show that the clustering effect and hierarchical classification learned by robust model benefits the task as well.

## 2 Related Work

### 2.1 Adversarial Attack and Defense

**Attack** Given a clean example $x$, its true label $y$, a DNN model $\mathcal{F}$ with parameters $\theta$ and its loss function $\mathcal{L}$, the goal of adversarial attack is to find an adversarial example $x'$ that is close to clean example on pixel level but can fool $\mathcal{F}$ to give an incorrect prediction. Fast Gradient Sign Method (FGSM) [13] adds perturbations on clean example $x$ by one step with step size $\epsilon$:

$$x' = x + \epsilon \text{sign}(\nabla_x \mathcal{L}(\mathcal{F}(x, \theta), y)). \tag{1}$$

Projected Gradient Descent (PGD) [23] adds perturbations on clean example $x$ by $K$ steps with smaller step size $\alpha$. After every step ($k$-th) attack, adversarial example is projected into the $\epsilon$-ball $\mathcal{B}_\epsilon(x)$ around clean example (function $\Pi$):

$$x^k = \Pi(x^{k-1} + \alpha \text{sign}(\nabla_x \mathcal{L}(\mathcal{F}(x^{k-1}, \theta), y))). \tag{2}$$

There are other attack techniques, such as white-box CW attack [6], black-box attacks [21, 3] and adaptive AutoAttack [9].

**Defense** Many adversarial defense techniques have been proposed since then, such as input pre-processing [19], defensive distillation [24], model compression [10], and adversarial training [23]. Among them, adversarial training variants are assumed to be most effective in comprehensive attack evaluations. They solve a min-max problem as:

$$\min_\theta \max_{x' \in \mathcal{B}_\epsilon(x)} \mathcal{L}(\mathcal{F}(x', \theta), y). \tag{3}$$

The inner maximum problem often generates adversarial examples by FGSM or PGD, while the outer minimum problem optimizes the worst-case loss. Some defense technique improves robustness through class-wise feature clustering [1, 30], which is different from our instance-wise feature clustering.

### 2.2 Linearity Exploration in Adversary

Beside attack or defense techniques, the cause and understanding of adversary are studied. The linearity is one essential perspective with no doubt. Previous studies attribute the existing of adversarial examples to the linearity of DNNs [13] and observe the adversarial transferability enhancement on more linear models [34, 14]. Some works study linearity from another perspective by focusing on the properties of weight layers, *e.g.*, their norms, variances and orthogonality. $L_p$ norm regularization works by pulling examples far away from the decision boundary, which comes as the side-effect of adversarial robustness [37, 18]. Spectral norm [7, 26] is useful in adversarial defense, which is defined as $\sigma_i = \frac{\|w_i x\|_2}{\|x\|_2}$ and is used to constrain the Lipschiz constant of a DNN $\mathcal{F}$ on $x$ (assuming Lipschiz continuous) $\|\mathcal{F}(x + \delta) - \mathcal{F}(x)\|_2 \leq \sigma \|x\|_2 \leq \prod \sigma_i \|x\|_2$. The weight scale shifting issue is also discussed in adversary [22], that is, scale of weights can be shifted between layers without changing the input-output function specified by the network, which could affect the capacity to regularize models. Then one weight scale shift invariant regularization is proposed and improves adversarial robustness. Moreover, the orthogonality could help improve generalization and adversarial robustness [5, 7] by inducing uncorrelated features. However, these experiments are all conducted on the original non-linear models instead of the linear components directly, which hinders a deeper understanding on linearity.

## 3 Observations on Linear Components

The robust models perform completely differently from non-robust ones especially under comprehensive adversarial attacks. However, the statistical regularity on linear components of these robust models is yet under little exploration. Different from related works, we explore linear components directly. To be specific, we extract a weight matrix expression of linearized sub-networks, estimating the linear propagation from inputs to outputs.

### 3.1 Extracting a Linear Weight Matrix

Given a DNN $\mathcal{F}$ composed of $L$ linear layers and non-linear components, the output $y = \mathcal{F}(\boldsymbol{x})$ could be expressed as

$$y = g_L(w_L \times g_{L-1}(w_{L-1} \times \ldots g_1(w_1 \times \boldsymbol{x} + b_1) \cdots + b_{L-1}) + b_L), \tag{4}$$

where $g_i$ is a series of non-linear components (*e.g.*, activation function ReLUs), $w_i$ and $b_i$ are the expressions of linear ones. Then a corresponding linear sub-network of this original one could be extracted with the same weights and architectures, yet it removes activation function (*e.g.*, ReLU layers) and batch normalization layers, and further replaces maximum pooling layers with average pooling layers. This output of linear network $y_{\mathrm{linear}} = \mathcal{F}_{\mathrm{linear}}(\boldsymbol{x})$ is expressed as

$$y_{\mathrm{linear}} = w_L \times (w_{L-1} \times \ldots (w_1 \times \boldsymbol{x} + b_1) \ldots + b_{L-1}) + b_L. \tag{5}$$

Thus in $\mathcal{F}_{\mathrm{linear}}$, each input $\boldsymbol{x}$ is multiplied by an instance-agnostic total weight

$$W = w_L \times w_{L-1} \times \ldots w_1 + \sum_{i,j} b_i \ldots \times w_j \ldots . \tag{6}$$

However, such total weight expression $W$ in DNNs is difficult to compute directly, because the multiplication of convolution layers is infeasible. Instead, we propose to compute $W$ by applying backward propagation on the linear sub-network. To be specific, given a random input $\boldsymbol{x}$ and a pre-trained DNN $\mathcal{F}$, $W$ is computed as the gradient propagated on linear sub-network $\mathcal{F}_{\mathrm{linear}}$ following Algorithm 1. For a non-linear model, as the activation part is example-wise, it is difficult to analyse the original network with non-linear components. In contrast, the linear component is generally applied on all examples and example-agnostic. So we extract a linear expression from the original non-linear model to approximate and analyse the model performance. Intuitively, although the linearized network has limited capacity, we think that the class-wise directions (the extracted linearized weight vectors) should still represent some class-wise directions, which holds a connection with the feature space of the non-linear models.

---

**Algorithm 1** Extracting the Linear Weight Matrix $W$

---

**Input:** A random input $\boldsymbol{x}$, a pre-trained DNN model $\mathcal{F}$ with $D_{\mathrm{output}}$ classes
**Output:** Weight matrix $W$
   Get the corresponding linear sub-network $\mathcal{F}_{\mathrm{linear}}$ from $\mathcal{F}$
   Conduct forward propagation as: $y_{\mathrm{linear}} = \mathcal{F}_{\mathrm{linear}}(\boldsymbol{x})$
   **for** $i$ in range($D_{\mathrm{output}}$) **do**
      $y_{\mathrm{linear}}[i].\mathrm{backward}()$
      $W[:, i] = \boldsymbol{x}.\mathrm{grad}$
   **end for**
   **return** $W$

---

### 3.2 Observation on Weight Clustering Effect

As indicated, though instance-wise performance varies on one specific DNN with non-linear components, it shares the same linear sub-network (*i.e.* parameters and linear structures) and thus the same implicit weight matrix $W$. As the implicit matrix $W$ could estimate linear output scores $y_{\mathrm{linear}}$ of any input $\boldsymbol{x}$ when forward propagating on the linear sub-network, it hints some class-wise linear amplifications. After extracting $W$ with shape $D_{\mathrm{input}} \times D_{\mathrm{output}}$, we further explore the correlation across classes by normalizing weight vectors in $W$ (to avoid the class-wise scale variance) and computing a correlation matrix $C$ following

$$C_{i,j} = \frac{W_i^T}{\|W_i^T\|_2} \times \frac{W_j}{\|W_j\|_2}. \tag{7}$$

In such fashion, the matrix $C$, whose shape is $D_{\mathrm{output}} \times D_{\mathrm{output}}$, has all elements in [-1, 1]. $C(i, j)$ represents for cosine value of two weight vectors corresponding to class $i$ and $j$ on linear sub-networks, which also represents for the correlation of class-wise weight vectors. If the element $C(i, j)$ is close to 1, it means that $i$-th and $j$-th class-wise weight vectors are strongly positive correlated in the linear weight space, which also demonstrates that class $i$ and $j$ are more likely to be positively related with

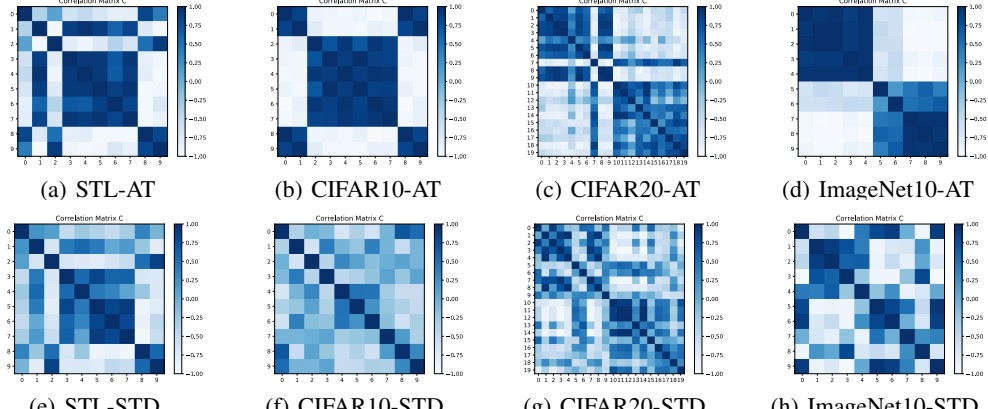

(a) STL-AT     (b) CIFAR10-AT     (c) CIFAR20-AT     (d) ImageNet10-AT

(e) STL-STD     (f) CIFAR10-STD     (g) CIFAR20-STD     (h) ImageNet10-STD

Figure 2: Correlation weight matrix $C$ of ResNet-18 on different data sets. 'AT' indicates adversarial training. 'STD' indicates standard training. Robust models tend to show weight clustering effect aligned with class hierarchy.

each other. In contrast, for the element $C(i,j)$ close to -1, it means that $i$-th and $j$-th class-wise weight vectors are strongly negative correlated in the linear weight space, which also demonstrates that class $i$ and $j$ are more likely to be negatively related with each other.

**Datasets and Setups** Beside Fig. 1, we show more clustering effect across data sets using STL-10 [8] and two reconstructed data sets CIFAR-20 and ImageNet-10, composed of 20/10 subclasses from 4/2 superclasses in CIFAR-100 and TinyImageNet [27]. The details of constructed data sets are in Appendix A. For robust models, we adversarially train ResNet-18 [15] using PGD-10 ($\epsilon = 8/255$ and step size $2/255$) with random start. The non-robust models are standard trained. Both models are trained for 200 epochs using SGD with momentum 0.9, weight decay 2e-4, and initial learning rate 0.1 which is divided by 10 at the 75-th and 90-th epoch.

**Result Analysis** We plot the correlation matrix $C$ in Fig. 2. Similar to Fig. 1 on CIFAR-10, the similar block clustering effect is observed on robust models and aligns well with class hierarchy. To be specific, in Fig. 2(a) and 2(e) on STL-10, fine labels (0,2,8,9) belong to 'Non-animal', others belong to 'Animal', and the matrix $C$ of robust model shows block clustering: values close to '+1' in the two superclasses (0,2,8,9) and (1,3,4,5,6,7), values close to '-1' across these two superclasses. Such phenomena hint at the fundamental connection of adversarial robustness with weight clustering effect. That is, subclasses in the same superclass are positively related to each other, which is in contrast with those in different superclasses. This hierarchical classification property is more semantic with data sets, demonstrating that robust model has a better representative feature extraction capacity and thus outperforms standard model in various tasks.

**Discussion** In order to analyse the connection of weight clustering effect on linearized models with their original models, we specifically take the linear model as an example whose linearized model is itself. We additionally train two linear models with initial learning rate 0.01 and 100 epochs by standard training or PGD-10 adversarial training. The detailed architecture of the model is shown in Appendix F. The final accuracies are listed in Table 1. Though the model capacity restricts their accuracies, the robust model still shows a clustering effect. Fig. 3 indicates that the clustering effect of linear weight matrix generally holds even on a simplified linear model.

Table 1: Accuracy (%) on CIFAR-10 of two linear models. 'AT'/'STD' indicate adversarial/standard training. The model capacity restricts their accuracies.

| Linear Model (CIFAR-10) | AT | STD |
|---|---|---|
| Clean ACC | 29.21% | 39.53% |
| PGD-20 ACC | 20.80% | 4.36% |

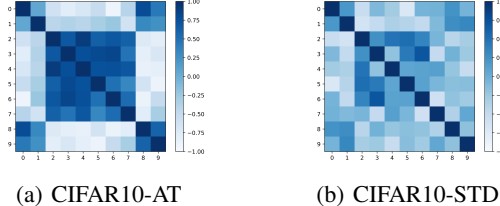

(a) CIFAR10-AT     (b) CIFAR10-STD

Figure 3: Correlation weight matrix $C$ of linear models on CIFAR-10. 'AT' model shows a clustering effect.

# 4 Explorations with Clustering Effect

A simple and direct idea is to apply the linear weight clustering effect as a penalty. However, the time cost and computational cost are too expensive. Based on our understanding that robust models learn better class hierarchy of one data set and thus better semantic features, we further add the penalty in feature space by evaluating a class-wise feature distance matrix $D$ of original non-linear DNNs on CIFAR-10. Given data pairs $(\boldsymbol{x}_i, y_i)$, their features $\boldsymbol{z}_i$, we compute the class-wise feature centers $\boldsymbol{Z}_j$ for class $j$ following

$$\boldsymbol{Z}_j = \text{mean } \boldsymbol{z}_i \cdot \mathbb{1}(y_i = j) \tag{8}$$

$F(a, b)$ represents the distance of class-wise feature centers $\boldsymbol{Z}_a$ and $\boldsymbol{Z}_b$, which is normalized with the largest class-wise distances for a better visualization.

$$F(a, b) = \frac{\|\boldsymbol{Z}_a - \boldsymbol{Z}_b\|_2}{\max\limits_{c, c \neq a} \|\boldsymbol{Z}_a - \boldsymbol{Z}_c\|_2}. \tag{9}$$

**Result Analysis** We adopt the same basic setting as Sec. 3.2, and find a similar clustering effect on CIFAR-10 in Fig. 4. The models are the same with those in Fig. 2. Robust models consistently show the better clustering effect compared with non-robust models. That is, the centers of subclasses in the same superclass lie closer to each other in feature space. The feature clustering effect also aligns well with class hierarchy. Such effect further confirms our understanding that robust models could extract more semantic and hierarchical features even in their original non-linear feature space.

**Further Clustering Enhancement** Given any unknown data set, we first extract its class hierarchy from a pre-trained robust model, *e.g.*, trained by adversarial training. To be specific, we compute matrices $W$ and $C$ following Sec. 3.2. We then formulate $C$ to an approximate matrix $C_{\text{op}}$ with all items set to '-1' or '+1', *i.e.*, in $C_{\text{op}}$, subclasses in the same superclass block have values '+1', else have values '-1'. With the extracted class hierarchy, a regularization loss is designed to minimize outputs within the same superclass. Take the data set CIFAR-10 for example, subclasses are divided into (0,1,8,9) and (2,3,4,5,6,7) in $C_{\text{op}}$. The regularization loss is defined in Eq. 10. The hierarchical instance-wise clustering training strategy is proposed following Algorithm 2.

$$
\begin{aligned}
F_0 &= \mathbb{E}_x \mathcal{F}(\boldsymbol{x})_i \cdot \mathbb{1}(i = 0, 1, 8, 9) \\
F_1 &= \mathbb{E}_x \mathcal{F}(\boldsymbol{x})_j \cdot \mathbb{1}(j = 2, 3, 4, 5, 6, 7) \\
\mathcal{L}_{\text{reg}}(\boldsymbol{x}) &= \sum_{i=0,1,8,9} \|\mathcal{F}(\boldsymbol{x})_i - F_0\|_2 + \sum_{j=2,3,4,5,6,7} \|\mathcal{F}(\boldsymbol{x})_j - F_1\|_2.
\end{aligned}
\tag{10}
$$

---

**Algorithm 2** Enhancing the Hierarchical Clustering Effect

---

**Input:** A random input $\boldsymbol{x}$, a pre-trained robust DNN model $\mathcal{F}$ (*e.g.*, on CIFAR-10)
**Output:** Robustness Enhanced Model $\mathcal{F}$
    Compute the weight correlation matrix $C$ following Algorithm 1 and Eq. 7
    Compute an approximate matrix $C_{\text{op}}$ with all items '+1/-1' based on $C$ and extract class hierarchy
    Compute regularization loss $\mathcal{L}_{\text{reg}}(\boldsymbol{x})$ with feature clusters following Eq. 10
    Retrain $\mathcal{F}$ from scratch using the original robust loss $\mathcal{L}(\boldsymbol{x}, y)$ (*e.g.*, adversarial training loss) added with $\mathcal{L}_{\text{reg}}$
    **return** $\mathcal{F}$

---

## 4.1 Domain Adaption Case

Recent study shows that robust models tend to perform better in downstream tasks, especially domain adaption with subpopulation shift [29]. However, further analyses are insufficient. We study from two perspectives, 1) verifying how robust models outperform in domain adaption with different settings and 2) exploring a further improvement by enhancing the clustering effect. Specifically, we randomly divide a hierarchical data set into source data and target data according to their subclasses. That is, source and target data share the same superclass yet different subclasses, dubbed subpopulation shift. In the training phase, we use fine labels of source data and map fine labels to coarse ones. In such a fashion, the model returns fine and coarse labels. Only coarse label is evaluated in the test phase as the target test data has different fine labels from source data. Both coarse and fine accuracies are

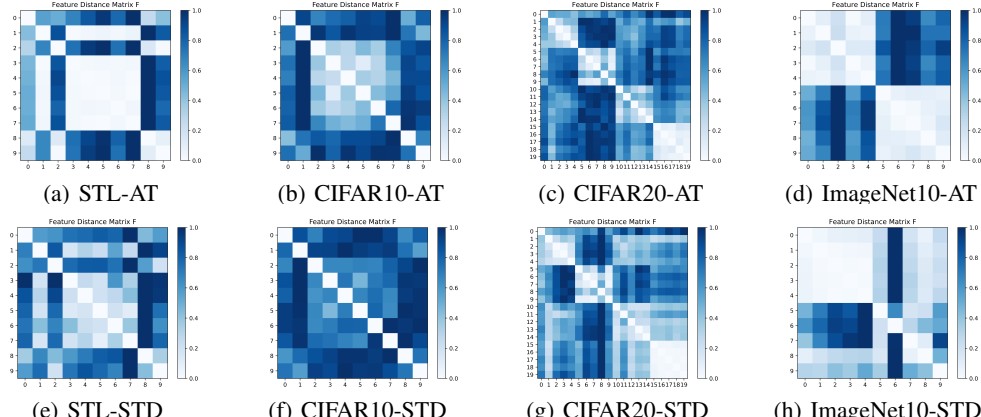

| (a) STL-AT | (b) CIFAR10-AT | (c) CIFAR20-AT | (d) ImageNet10-AT |
|---|---|---|---|
| (e) STL-STD | (f) CIFAR10-STD | (g) CIFAR20-STD | (h) ImageNet10-STD |

Figure 4: Feature distance matrix $F$ of ResNet-18 on different data sets. 'AT' indicates adversarial training. 'STD' indicates standard training. Robust models tend to show feature clustering effect aligned with class hierarchy, which is similar to $C$.

evaluated on source domain. We enhance the feature clustering effect by a penalty loss defined in Eq. 10. To better utilize hierarchical features extracted by pre-trained models, we also finetune parameters on target data to achieve a further improvement. Different models, different constructed data sets and different adversarial training settings (*i.e.* adversarial robustness levels) are evaluated and analyzed.

#### 4.1.1 Different Models

**Datasets and Setups** The data set CIFAR-100 is composed of 20 superclasses, and each superclass is composed of 5 subclasses. Based on that, we construct our own data sets by randomly sampling from CIFAR-100, *e.g.*, our default CIFAR-20 is composed of 20 subclasses from 4 superclasses. The details of all constructed data sets are shown in Appendix A. We adversarially train ResNet-18, DenseNet-121 [17], and AlexNet [20] with PGD. If not specifically mentioned, PGD-5 is adopted (5 steps, step size 1/255, $\epsilon$=2/255). The robust/non-robust ResNet-18 and DenseNet-121 are trained with SGD [25] and a initial learning rate 0.1. The robust/non-robust AlexNet are trained with SGD and a initial learning rate 0.01. The learning rates decay with a factor of 0.1 at the 75 and 90 epoch. Other hyper-parameters are the same with Sec. 3.2. Moreover, we finetune the pre-trained models on target data for 20 epochs with a learning rate 0.01.

**Results Analysis** We evaluate the models on our default CIFAR-20. As shown in Table 2, when comparing the coarse accuracies on target and source data, robust models tend to decline less than non-robust ones. Take pre-trained ResNet-18 for example, the coarse accuracy on robust model directly declines by 5.44% (from 87.94% to 82.50%) but declines by 10.18% (from 90.93% to 80.75%) on non-robust model. When considering finetuned models, the superiority of robust models outstands. For example, the coarse accuracy on finetuned robust ResNet-18 increases by 6.46% (from 87.94% to 94.40%) while increases by 3.32% (from 90.93% to 94.25%) on finetuned non-robust ResNet-18. When applying the clustering penalty defined in Eq. 10, coarse accuracy is improved by 2.25% on robust ResNet-18 (from 82.50% to 84.75%) even without finetuning. Experimental results demonstrate that features extracted by robust models have a non-trivial impact on domain adaptation. Such superiority keeps consistent when finetuning with the feature centers of target training data. Thus our understanding is confirmed that a better hierarchical classification learned by robust models benefits a lot in downstream domain adaption task.

#### 4.1.2 Different Data Sets

**Datasets and Setups** For CIFAR-100, we randomly select 4 to 6 superclasses, select 4 subclasses from each superclass for training and apply the left one for test. We also select 5 superclasses, select 3 subclasses from each superclass for training and apply the left 2 subclasses for test. When selecting 4 superclasses and 4 subclasses, we typically design two extreme data set construction methods, *i.e.* the most different and similar data set. The 'different' 4 superclasses are selected as index 1, 2, 5, 14 (fish, flowers, household electrical devices, people). The 'similar' 4 superclasses are selected as index 8, 11, 12, 16, which are all animals. The reconstructed data set ImageNet-20 is composed of

Table 2: Accuracy (%) on CIFAR-20 across various models. 'Coarse' and 'Fine' indicate the ground-truth labels are coarse or fine respectively. 'FT' indicates the pre-trained models are finetuned on target data. 'R' indicates robust models (adversarial training). 'NR' indicates non-robust models (standard training). '+C' indicates robust models with an enhanced clustering effect defined in Eq. 10.

| | | Source Domain | | Target Domain | |
|---|---|---|---|---|---|
| | | Coarse | Fine | Coarse | Coarse-FT |
| AlexNet | NR | 85.50 | 65.37 | 71.75 | 86.70 |
| | R | 84.75 | 64.50 | **72.75** | **87.50** |
| | R+C | 90.68 | 65.56 | **75.25** | **89.25** |
| ResNet-18 | NR | 90.93 | 70.62 | 80.75 | 94.25 |
| | R | 87.94 | 68.75 | **82.50** | **94.40** |
| | R+C | 88.25 | 69.62 | **84.75** | **94.75** |
| DenseNet-121 | NR | 92.31 | 73.56 | 77.25 | 95.50 |
| | R | 90.18 | 72.87 | **81.00** | **95.75** |
| | R+C | 91.37 | 73.87 | **85.25** | **96.50** |

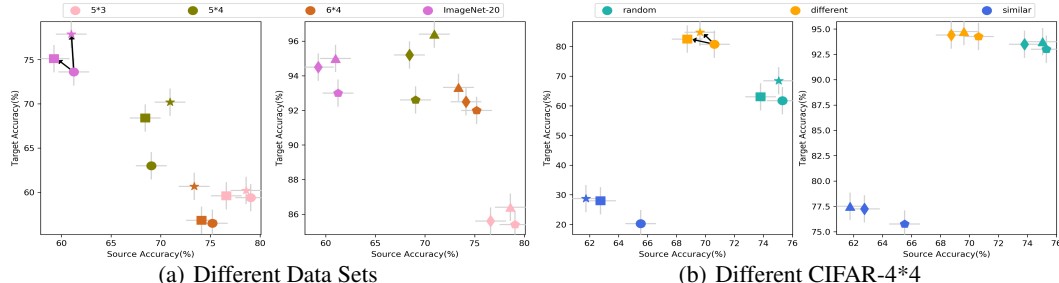

(a) Different Data Sets                    (b) Different CIFAR-4*4

Figure 5: Comparisons of target accuracy across different data sets. Different colors indicate different data set construction methods. Different shapes indicate different training methods, *i.e.* adversarial or standard training. Circle indicates non-robust model (NR), square and start indicate robust model without/with an enhanced clustering effect (R/R+C) respectively. Pentagon, diamond and triangle indicate finetuning NR/R/R+C models respectively. The legend in format $A * B$: $A$ indicates the number of selected superclasses, $B$ indicates the number of selected subclasses from each superclass for training. The points located in left and upper represent for better performances as they have lower source accuracies yet higher target accuracies.

20 subclasses from 4 superclasses in TinyImageNet. The details of datasets are in Appendix A. We conduct experiments on ResNet-18 in this section.

**Results Analysis** As shown in Fig. 5, we find that experimental results on different data sets show different yet consistent performances: robust models show great superiority on those more hierarchical data sets, which can be boosted with an enhanced clustering effect. In Fig. 5(a), our constructed data sets from CIFAR-100 and TinyImageNet show superiority with robustness and an enhanced clustering. Moreover, a better class hierarchy of data set enhances domain adaption tasks. For example, orange points (*i.e.* the 'different' data set) in both two subfigures in Fig. 5(b) tend to present lower source accuracies yet higher target accuracies, especially compared with lightseagreen points (*i.e.* the 'random' data set). For 'similar' data sets, because the class hierarchy is not obvious, target accuracy drops a lot. However, the royalblue start point (*i.e.* 'R+C' version model trained on 'similar' data set) still achieves the highest target accuracies among three models trained on 'similar' data set. We show the specific numerical results in Appendix C.

### 4.1.3 Different Robustness Settings

**Datasets and Setups** We evaluate ResNet-18 with different robustness settings (*i.e.* different PGDs in adversarial training) on random CIFAR-4⋆4 defined in Appendix A. The PGD details are given in Appendix B.

**Results Analysis** As shown in Fig. 6, we could observe a trade-off between adversarial robustness and accuracies on target data. For example, in the polylines of both two subfigures, purple points show higher accuracies on target data than brown ones. However, brown points are of better robustness as their accuracies on source data are much lower. Royalblue points show the relatively bad performance

on target data yet the best performance on source data. The phenomena demonstrate a trade-off between robustness and domain adaption performances. The better performance on target domain could only be achieved with both robustness and moderate accuracies on source data. Furthermore, robust model with an enhanced clustering effect has significantly results than the original robust model. We show the specific numerical results in Appendix C. In addition, the time costs results with different robustness settings are shown in Appendix D.

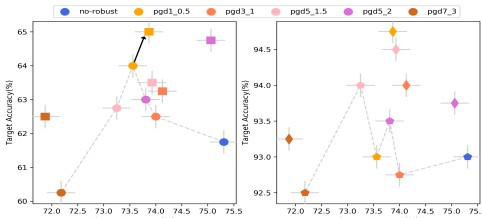

Figure 6: Comparisons of target accuracy across different adversarial robustness levels (*i.e.* different adversarial training settings). Different colors indicate different PGD attack settings in adversarial training. Circle and square indicate robust model without/with an enhanced clustering effect (R/R+C). Pentagon and diamond indicate finetuning R/R+C model respectively. The points located in left and upper represent for better performances as they have lower source accuracies yet higher target accuracies.

## 4.2 Adversarial Case

We further conduct extensive experiments to explore such an intriguing clustering effect along with other adversarial phenomena. The experiments are designed from two perspectives, 1) adversarial robustness enhancement and 2) attack confusion matrix study. The results show that adversarial robustness could be improved by enhancing the hierarchical clustering effect, and attack success rates are highly related to class hierarchy as well.

### 4.2.1 Adversarial Robustness Enhancement

**Datasets and Setups** We adversarially train a robust ResNet-18 on CIFAR-10. The basic training setting is the same with Sec. 3.2. The PGD-20 attack is applied with $\epsilon = 8/255$ and step size 0.003.

**Results Analysis** Although the main exploration of clustering effect is studied not only in robustness improvement, we conduct comprehensive robustness evaluations following the state-of-the-art attack strategy with an adaptive AutoAttack [9]. We find that the clustering enhancement actually boosts adversarial robustness in Table 3 for both best and last checkpoints. For black-box attacks, the surrogate model is a standard trained VGG-16. $\mathcal{N}$Attack is conducted on 1,000 random test data and with a maximum query 20,000. Note that, the improvement against AutoAttack is 1.31%, which shows that the clustering alignment with coarse labels learned by robust models could indeed boost adversarial robustness. An interesting phenomenon is that when enhancing clustering effect, both natural and robust accuracies are improved because the clustering effect improves feature representation. The natural accuracy by standard training improves as well (from 92.75% to 93.54%) with a clustering effect. Therefore, the trade-off between adversarial robustness and natural accuracy still exists. In addition, we count the training time of two models (AT and AT+**C**). The AT model costs 191.20s per epoch, while the AT+**C** model costs 193.66s, using one GPU 1080X with batch size 128 of ResNet-18 on CIFAR-10. The time cost slightly increases with the clustering regularization, which is negligible. More results are shown in Appendix E.

Table 3: Robustness (accuracy (%) on various attacks) on CIFAR-10, based on the best/last checkpoint of ResNet-18. 'AT' indicates adversarial training. '**+C**' indicates applying our hierarchical instance-wise clustering training strategy. The best results are **boldfaced**.

| Defense | | White-box | | | | Black-box | | | Adaptive |
|---|---|---|---|---|---|---|---|---|---|
| | | Natural | FGSM | PGD-20 | CW$_\infty$ | PGD-20 | CW$_\infty$ | $\mathcal{N}$Attack | AutoAttack |
| AT | Best | 84.20 | 63.32 | 50.12 | 50.97 | 81.80 | 84.80 | 48.21 | 46.58 |
| | Last | 84.27 | 60.46 | 46.50 | 48.97 | 79.13 | **79.87** | 45.05 | 41.90 |
| AT+**C** | Best | **85.43** | **64.11** | **52.54** | **52.72** | **82.27** | **85.22** | **50.51** | **47.21** |
| | Last | **85.53** | **62.11** | **47.78** | **49.95** | **80.21** | 78.98 | **47.51** | **43.21** |

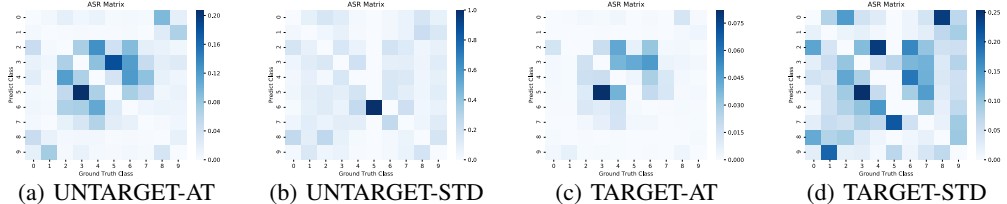

|(a) UNTARGET-AT|(b) UNTARGET-STD|(c) TARGET-AT|(d) TARGET-STD|

Figure 7: Attack confusion matrix $M$ under different settings. 'AT'/'STD' indicate adversarial/standard training. Robust models tend to show clustering effect aligned with class hierarchy.

### 4.2.2 Attack Confusion Matrix

To further analyze how class hierarchy matters in adversary, we evaluate the attack confusion matrix $M$ of adversarial examples in Fig. 7, whose shape is $D_{\text{output}} \times D_{\text{output}}$. $M(i, j)$ is defined as attack success rate (ASR) of adversarial examples being attacked from the ground-truth label $i$ to the target label $j$.

**Datasets and Setups** We train robust and non-robust ResNet-18 on CIFAR-10. The basic setting is the same with Sec. 3.2. The PGD-20 attack is applied with $\epsilon = 8/255$ and step size 0.003. For the target attack, target labels are set as the other 9 classes except for the ground-truth one.

**Results Analysis** Experimental results show that the attack confusion matrix $M$ performs in another blocking pattern especially on robust models, which is well-aligned with class hierarchy as well as the weight correlation matrix $C$ and the feature distance matrix $F$. That is, the difficulty in attacking robust models shows distinguishing performance for subclasses in or out of the same superclass. For example, against untarget PGD-20 attack in Fig. 7(a), the ASRs on adversarially trained ResNet-18 show that classes in (2,3,4,5,6,7) are easier to be attacked from one to other. However, the ASRs could decline to almost zero from one superclass (0,1,8,9) to the other (2,3,4,5,6,7). Different superclasses are harder to attack especially on robust models, demonstrating that the hierarchical classification capacity is enhanced by adversarial robustness. Such phenomenon is also consistent with our hierarchical clustering observations on robust models.

## 5 Conclusions

In this paper, we provide a novel view of linear model exploration by extracting an implicit linear matrix expression. Then we surprisingly find an intriguing clustering effect on adversarially robust models, which is well-aligned with class hierarchy. Based on such observations, we first give an insightful explanation and understanding of robust models, which have a better capacity in extracting more representative features and semantic information. Extensive experiments show that enhancing the hierarchical clustering effect not only improves model robustness, but also benefits other downstream domain adaption task. Overall, our observations and findings motivate a deeper understanding on adversarial robustness.

## Acknowledgement

Yisen Wang is partially supported by the National Natural Science Foundation of China under Grant 62006153, and Project 2020BD006 supported by PKU-Baidu Fund. Shu-Tao Xia is supported in part by the National Key Research and Development Program of China under Grant 2018YFB1800204, the National Natural Science Foundation of China under Grant 62171248, the R&D Program of Shenzhen under Grant JCYJ20180508152204044.

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
