# A  Data Sets Construction

The details of default CIFAR-20 and ImageNet-10 in our paper are in Table 4. For CIFAR-20, we sample 4 superclasses with all their subclasses randomly. For ImageNet-10, we sample 2 superclasses with all their subclasses randomly.

Table 4: The details of CIFAR-20 and ImageNet-10. We list coarse labels, in which all fine labels are included.

| Data sets | Coarse Labels | | | |
|---|---|---|---|---|
| CIFAR-20 | fish | flowers | household electrical devices | people |
| ImageNet-10 | animal | non-animal | | |

The details of a series of constructed data sets are in Table 5. As we study the domain adaption tasks on such data sets, we also divide them into training set and test set according to their subclasses randomly.

Table 5: The construction of CIFAR sub-datasets and TinyImageNet sub-datasets. The legend in format $A \star B$: $A$ indicates the number of selected superclasses, $B$ indicates the number of selected subclasses from each superclass for training. 'Diff' indicates the constructed CIFAR-20 with different coarse labels. 'Sim' indicates the constructed CIFAR-20 with similar coarse labels. 'Rand' indicates the constructed CIFAR-20 with random coarse labels. 'I-20' indicates the constructed ImageNet-20 from TinyImageNet. We list the coarse labels of sub-datasets.

| Data sets | Coarse Labels | | | | | |
|---|---|---|---|---|---|---|
| Diff | fish | flowers | household electrical devices | people | | |
| Sim | large carnivores | large omnivores and herbivores | medium-sized mammals | small mammals | | |
| Rand | fruit and vegetables | household electrical devices | household furniture | insects | | |
| 5 ⋆ 3 | fruit and vegetables | household electrical devices | household furniture | insects | large carnivores | |
| 5 ⋆ 4 | fruit and vegetables | household electrical devices | household furniture | insects | large carnivores | |
| 6 ⋆ 4 | fruit and vegetables | household electrical devices | household furniture | insects | large carnivores | large man-made outdoor things |
| I-20 | animals | houses and landscapes | foods | fruits | | |

# B  Attack Settings

The PGD attack settings used in Sec 4.1.3 are in Table 6. We adopt the following PGDs in adversarial training for diverse robustness levels.

Table 6: The hyper-parameters of various PGD attack settings.

| Attack | Steps | Epsilon ($\epsilon$) | Step size |
|---|---|---|---|
| PGD7_3 | 7 | 3/255 | 1/255 |
| PGD5_2 | 5 | 2/255 | 1/255 |
| PGD5_1.5 | 5 | 1.5/255 | 0.5/255 |
| PGD3_1 | 3 | 1/255 | 0.5/255 |
| PGD1_0.5 | 1 | 0.5/255 | 0.5/255 |

## C  More Results for Domain Adaption

Besides the numerical results for domain adaption experiments across different models in Sec. 4.1.1, we add more numerical results across different data sets and across different robustness levels in Table 7 and Table 8 respectively. In addition, we also evaluate on ImageNet-20 in Table 9. The models trained with an enhanced clustering effect ('+C') show consistently better performances with/without finetuning ('FT') on target data, which is the same with Sec. 4.1.

Table 7: Accuracy (%) of ResNet-18 on various data sets. 'Coarse' and 'Fine' indicate the ground truth labels are coarse or fine respectively. 'FT' indicates the pre-trained models are finetuned on target data. 'R' indicates robust models (by adversarial training). 'NR' indicates non-robust models (by standard training). '+C' indicates the robust models with penalty following Eq. 10. The legend in format $A \star B$: $A$ indicates the number of selected superclasses, $B$ indicates the number of selected subclasses from each superclass for training.

|  |  | Source Domain | | Target Domain | |
|---|---|---|---|---|---|
|  |  | Coarse | Fine | Coarse | Coarse-FT |
| $5 \star 3$ | NR | 87.20 | 79.06 | 59.40 | 85.40 |
|  | R | 84.66 | 76.60 | **59.60** | **85.50** |
|  | R+C | 86.86 | 78.60 | **60.20** | **86.40** |
| $4 \star 4$ | NR | 87.87 | 75.31 | 61.75 | 93.00 |
|  | R | 85.62 | 73.81 | **63.00** | **93.50** |
|  | R+C | 86.18 | 75.06 | **64.75** | **93.75** |
| $5 \star 4$ | NR | 89.00 | 69.05 | 63.00 | 92.60 |
|  | R | 87.40 | 68.45 | **68.40** | **95.20** |
|  | R+C | 89.05 | 70.95 | **70.20** | **96.40** |
| $6 \star 4$ | NR | 87.29 | 75.20 | 56.50 | 92.00 |
|  | R | 84.54 | 74.12 | **56.83** | **92.50** |
|  | R+C | 85.37 | 73.37 | **60.66** | **93.33** |
| Different | NR | 90.93 | 70.62 | 80.75 | 94.25 |
|  | R | 87.94 | 68.75 | **82.50** | **94.40** |
|  | R+C | 88.25 | 69.62 | **84.75** | **94.75** |
| Similar | NR | 76.87 | 65.52 | **30.25** | 75.75 |
|  | R | 74.00 | 62.75 | 28.00 | **77.25** |
|  | R+C | 74.81 | 61.56 | 28.75 | **77.50** |

## D  Time Costs

We count the time costs with/without our clustering enhancement in Table 10. The additional time cost is negligible. Our clustering training strategy is well acceptable.

## E  More Robustness Evaluations

For a more comprehensive robustness evaluation, we apply additional attack methods with multiple run times. To be specific, we conduct the robustness evaluation on the best checkpoint for 5 times, using the following attack methods. The results are shown in Table 11.

## F  A Linear Model Example

The architecture of linear model is given as follows.

Table 8: Accuracy (%) of ResNet-18 on CIFAR-4⋆4 across various robustness levels. 'Coarse' and 'Fine' indicate the ground truth labels are coarse or fine respectively. 'FT' indicates the pre-trained models are finetuned on target data. 'R' indicates robust models (by adversarial training). 'NR' indicates non-robust models (by standard training). '+**C**' indicates the robust models with penalty following Eq. 10.

| | | Source Domain | | Target Domain | |
|---|---|---|---|---|---|
| | | Coarse | Fine | Coarse | Coarse-FT |
| PGD7_3 | NR | 87.87 | 75.31 | 61.75 | 93.00 |
| | R | 85.18 | 72.18 | 60.25 | 92.50 |
| | R+**C** | 85.31 | 71.87 | **62.50** | **93.25** |
| PGD5_2 | NR | 87.87 | 75.31 | 61.75 | 93.00 |
| | R | 85.62 | 73.81 | **63.00** | **93.50** |
| | R+**C** | 86.18 | 75.06 | **64.75** | **93.75** |
| PGD5_1.5 | NR | 87.87 | 75.31 | 61.75 | 93.00 |
| | R | 85.62 | 73.25 | **62.75** | **94.00** |
| | R+**C** | 86.87 | 73.93 | **63.50** | **94.50** |
| PGD3_1 | NR | 87.87 | 75.31 | 61.75 | 93.00 |
| | R | 87.43 | 74.00 | **62.50** | 92.75 |
| | R+**C** | 87.68 | 74.12 | **63.25** | **94.00** |
| PGD1_0.5 | NR | 87.87 | 75.31 | 61.75 | 93.00 |
| | R | 85.31 | 73.56 | **64.00** | 93.00 |
| | R+**C** | 87.00 | 73.87 | **65.00** | **94.75** |

Table 9: Accuracy (%) on ImageNet-20 of ResNet-18. 'Coarse' and 'Fine' indicate the ground truth labels are coarse or fine respectively. 'FT' indicates the pre-trained models are finetuned on target data. 'R' indicates robust models (by adversarial training). 'NR' indicates non-robust models (by standard training). '+**C**' indicates the robust models with penalty following Eq. 10.

| ResNet-18 | Source Domain | | Target Domain | |
|---|---|---|---|---|
| | Coarse | Fine | Coarse | Coarse-FT |
| NR | 86.37 | 61.25 | 73.61 | 93.00 |
| R | 84.37 | 59.25 | **75.11** | **94.50** |
| R+**C** | 87.62 | 61.00 | **77.88** | **95.00** |

Table 10: The time costs of various PGD attack settings. 'AT' indicates robust models by adversarial training. '+**C**' indicates the robust models with penalty following Eq. 10. The additional time cost is negligible.

| Time Costs | PGD7_3 | PGD5_2 | PGD5_1.5 | PGD3_1 | PGD1_0.5 |
|---|---|---|---|---|---|
| AT | 25.45 | 21.43 | 21.46 | 17.44 | 13.42 |
| AT+**C** | 25.96 | 21.83 | 21.57 | 17.64 | 13.71 |

Table 11: Accuracy (%) on CIFAR-10, ResNet-18. We test on the best checkpoint and run for 5 times. 'AT' indicates robust models by adversarial training. '+**C**' indicates the robust models with penalty following Eq. 10. The clustering enhanced model ('+**C**') shows consistently better performances.

| | PGD-20 | DeepFool | JSMA | EAD |
|---|---|---|---|---|
| AT | 50.12±0.34 | 61.63±0.12 | 92.40±0.10 | 57.20±1.90 |
| AT+**C** | **52.54±0.12** | **62.56±0.35** | **93.45±0.35** | **58.60±0.80** |

```python
class Expression(nn.Module):
    def __init__(self, func):
        super(Expression, self).__init__()
        self.func = func

    def forward(self, input):
        return self.func(input)

class Model(nn.Module):
    def __init__(self, i_c=3, n_c=10):
        super(Model, self).__init__()

        self.conv1 = nn.Conv2d(i_c, 32, 5, stride=1, padding=2, bias=True)
        self.pool1 = nn.AvgPool2d((2, 2), stride=(2, 2), padding=0)

        self.conv2 = nn.Conv2d(32, 64, 5, stride=1, padding=2, bias=True)
        self.pool2 = nn.AvgPool2d((2, 2), stride=(2, 2), padding=0)

        self.flatten = Expression(lambda tensor: tensor.view(tensor.shape[0], -1))
        self.fc1 = nn.Linear(8 * 8 * 64, 1024, bias=True)
        self.fc2 = nn.Linear(1024, n_c)

    def forward(self, x_i):
        x_o = self.conv1(x_i)
        x_o = self.pool1(x_o)

        x_o = self.conv2(x_o)
        x_o = self.pool2(x_o)

        x_o = self.flatten(x_o)
        x_o = self.fc1(x_o)

        self.train()
        return self.fc2(x_o)
```

Figure 8: The structure of our linear model.