# OpenReview forum: "Clustering Effect of Adversarial Robust Models"
_NeurIPS.cc/2021/Conference — NeurIPS 2021 Spotlight_

### Official Review · Reviewer_7cFv · 2021-07-13

**Rating:** 5
**Confidence:** 5

**Summary:**

The paper analyses adversarially robust models in terms of their clustering behavior with respect to the learned weights as well as with respect to the class-wise feature representations. For this purpose, the CNNs are linearized, i.e. non-linearities (ReLUs) are removed from the trained network such that it can be represented by a single weight matrix W, where each column corresponds to one class. The paper comes to the conclusion that hierarchical clustering of these representations is an indicator for adversarial robustness. Consequently, it formulates a regularization term which encourages such clustering behavior. The resulting models are evaluated in terms of adversarial robustness and in the context of domain adaptation.

**Limitations And Societal Impact:**

The limitations and societal impact are not well discussed in the paper. With respect to the societal impact, I would assume that this is tolerable as the paper makes a conceptual contribution to the analysis of adversarial robustness, probably without a direct societal impact. Potential benefit would be, if the model works in practice , that CNNs could be applied in more safety relevant applications.
With respect to the limitations, please refer to the weaknesses I listed above. The model needs a significantly larger training time that the basic adversarially robust model. A comparison to the state of the art is missing and it is unclear whether the relative improvement would translate to a better model.


**Main Review:**

The paper makes an interesting observation in terms of analyzing adversarially robust models. While adversarial robustness is classically hard to grasp besides when evaluating against adversarial attacks, the proposed hierarchical clustering can be directly measured and implemented into a loss in a straight forward way. Therefore, the paper offers in my understanding an interesting new perspective onto the issue.
That said, several aspects are not clear in the paper.

From a theoretical perspective, it remains unclear why the analysis of the linearized network should extend to the original network with non-linearities. However, is this correlation holds, I wonder, from a practical perspective, why the feature space analysis is not executed on the linearized network as well. More importantly, in Figure 3, we see the evaluation of several trained models in terms of their linearized weight clustering. However, in Figure 4, only one model is shown in terms of feature clustering. The clustering effect appears to be much weaker here. What would the same evaluation look like for the other models in Fig.3? Also, in Fig 3. what are the accurcies on test/train/val set for the 8 models? Are these in a comparable range for each pair of models?

The experiments show that the additional proposed clustering loss can boost the robustness in terms of accuracy under attack for different attack methods, including the adaptive autoAttack method as well as a black-box attack. Similarly, the results on domain adaptation can be improved. Yet I object to the timing analysis in lines 213, 214. The model with the proposed loss only needs slightly more time to train than the original model with adversarial training. Yet, the entire model has to be fully trained before, in order to determine the clusters to be used in the regularization. Thus, the training time of the adversarially robust model is increased by more than a factor of 2.
The results in Tab 1 indicate that the accuracy under attack improves with the proposed procedure. Yet, the reported clean accuracy for these models is increased in a similar range (1.2 - 1.3 %) as well. So it is unclear whether the increased accuracy under attack might be caused by better training in general. The drop between clean accuracy and accuracy under attack remains about constant. Therefore, these results are not convincing to me.
The results in the domain adaptation task are more convincing. Yet, they are not well motivated in the context of the paper.
In neither of the evaluations, the proposed approach is compared to the state-of-the-art, but only relative improvements are reported. Would they also hold for the SotA model in terms of adversarial robustness such as:
https://arxiv.org/abs/2010.00467
https://arxiv.org/abs/2002.08619
https://arxiv.org/abs/2011.11164
https://arxiv.org/abs/2002.11242
etc... ?


The paper is overall well structured but the language is sometimes hard to understand. (e.g. line 48,49: D_input represents for the dimension...)


Minor:
Intro, line29: if you say "on the other hand", there should be "on the one hand" somewhere before
please define \delta in eq. 4


**Time Spent Reviewing:**

3

---

> ### Author Response · Authors · 2021-08-10
> **Response to Reviewer 3**
>
> Thanks for your suggestions and comments.
>
> **Q1.** Why can the weight extracted by the linearized network represent the original network?  Why not calculate the feature distance matrix in the linearized networks?
>
> **A1.** Thanks for your nice suggestion. Overall, we try to show a connection between the linearized weight space with the feature space.
> Firstly, we can simply start from a linear model, where $f(x)=k \times x$. Then the connection directly holds for weight vectors and feature space. We add the experiments on linear models. The clustering effect exists on robust models (we show the weight matrix figures in the following link: https://www.dropbox.com/sh/yljxu4rnj9j47xc/AACZDUk1dq4mibx2Bi-Ucnf9a?dl=0). However, the robust linear model on CIFAR-10 achieves a natural accuracy of 29.21\% in contrast with non-robust linear model 39.53\%. The accuracies are restricted by the linear model capacity.
> Secondly, as for a non-linear network, the proposed weight extraction method is universal on the whole dataset. The extracted weight vectors represent for some linearized class-wise weight directions. In addition, we explore the feature space and linearized weight space by some class-wise matrix. Specifically, we compare the correlation weight matrix $C$ of the linear sub-network with the feature distance matrix $E$ of the non-linear original network and find they show the similar clustering effect aligned with class hierarchy, which is shown in Figure 3 and the link. The figures reveal a nontrivial connection with matrices $C$ and $E$. They show the similar clustering effect. So the connection holds from linear sub-networks to original DNNs.
>
> **Q2.** Figure 2 does not show the effect of the feature distance matrix on other data sets. The results of the train/test/val data set in Figure 3.
>
> **A2.** We add new figures and results. We shown the feature distance matrix figures in the following link: https://www.dropbox.com/sh/yljxu4rnj9j47xc/AACZDUk1dq4mibx2Bi-Ucnf9a?dl=0.
> The figures show that robust models show consistent clustering effect compared with non-robust models. We also show the results of the training/test data set.
>
> | Models       |     | STL     | CIFAR-10 | CIFAR-20 | TinyImageNet-10 |
> | ------------ | --- | ------- | -------- | -------- | --------------- |
> | Training ACC | NR  | 91.14\% | 97.14\%  | 99.61\%  | 94.56\%         |
> |              | R   | 85.14\% | 93.17\%  | 95.92\%  | 91.31\%         |
> | Test ACC     | NR  | 78.71\% | 86.98\%  | 65.55\%  | 61.25\%         |
> |              | R   | 66.13\% | 84.20\%  | 60.10\%  | 59.25\%         |
>
> **Q3.** The timing analysis in lines \#213-\#214.
>
> **A3.** Thanks for your detailed reading. First, in our experimental settings, we assume that we do not know any category information of the dataset before training the network. So we need to train the network to get the weight correlation matrix $C$. But in many cases, we can get the category information in advance and manually design the weight correlation matrix $C$, then there is no need to pre-train the network first. Secondly, the re-training can be replaced by fine-tuning, in this case, our training time will not be 2 times than baselines.
>
> **Q4.** How effective is '+C' on other sota models?
>
> **A4.** We add more results on TRADES and FAT here. Our method brings a general improvement. We conduct experiments following their default settings. In fact, our methods worked from an understanding perspective and did not aim to be a new defense method, so we did not add many defense baselines.
>
> | ResNet-18s (CIFAR-10) | TRADES  | TRADES+C(Ours) | FAT     | FAT+C(Ours) |
> | --------------------- | ------- | -------------- | ------- | ----------- |
> | ACC against PGD-20    | 52.80\% | 53.71\%        | 43.76\% | 44.53\%     |

---

> > ### Comment · Reviewer_7cFv · 2021-08-16
> > **Thank you for the additional results and discussion**
> >
> > Dear Authors,
> > thank you for the additional experiments. The is especially appreciated since the available time was so short.
> > I will increase my score by 1 point.
> > Yet, I still think the paper should address tow remaining issues:
> >
> > W.r.t. point 1, I agree that the linearized network has limited capacity so the evaluation can not be particularly strong. Yet, I still don't understand the theoretic argumentation of why the clustering after the post hoc linearization should be characteristic since this linearization severely changes the behavior of the network. I miss a theoretical or even intuitive explanation for this point.
> >
> > W.r.t. point 4: The differences between the robustness improvements are very small, maybe even within the standard deviation when training seeds are varied. I would be careful with these results as no confidence intervals are provided.

---

> > > ### Author Response · Authors · 2021-08-18
> > > **Additional Responses**
> > >
> > > **Q1.** Intuitive explanation for the linear extraction.
> > >
> > > **A1.** Thanks for your inspired comments. We would like to give an intuitive explanation.
> > > Firstly, for a linear model $\mathcal{F}$, where $\mathcal{F}(x)=K \times x$, the linearized weight vectors $K$ can represent some class-wise directions.
> > > For example, the feature centers $\bar{f}$ (for class 0) should be $\bar{f}=k \times \bar{x}$ for $y=0$, where $\bar{f}$ and $\bar{x}$ represent the mean values. And the classification results (for class 0) should be $\bar{\mathcal{F}}(x) = \bar{f} \times fc = K \times \bar{x}$ for $y=0$, where $\bar{\mathcal{F}}(x)$ represents the mean value and $fc$ represents the fully-connection layer. $fc, K$ are example-agnostic.
> > > Therefore, there exists some direct and strong connection between linearized weight vectors $K$ and feature centers $\bar{f}$.
> > > Experimental results (in our manuscript Figures 3/4 and Rebuttal figure links) have also shown the consistent yet slightly different clustering performances of weight matrix $C$ and feature matrix $E$, as $\bar{f}$ is also influenced by the $\bar{x}$.
> > >
> > > Then for a non-linear model, as the activation part is example-wise, it is difficult to analyse the original network with non-linear components. In contrast, the linear component is generally applied on all examples and example-agnostic. So we extract a linear expression from the original non-linear model to approximate and analyse the model performance.
> > > Intuitively, we think that the class-wise directions ($\textit{i.e.}$, the extracted linearized weight vectors) should still represent some class-wise directions, which holds a connection with the feature space of the non-linear models.
> > >
> > > **Q2.** The improvement might be within the standard deviation.
> > >
> > > **A2.** We conduct 5 more run times on the models reported in Q4. The results are also listed as follows. In fact, we apply additional attack methods with multiple run times in Appendix Table 8. So the improvement is not due to one random run.
> > >
> > > | ResNet-18s (CIFAR-10) | TRADES           | TRADES+C(Ours)   | FAT              | FAT+C(Ours)      |
> > > | --------------------- | ---------------- | ---------------- | ---------------- | ---------------- |
> > > | ACC against PGD-20    | 52.98$\pm$0.34\% | 53.68$\pm$0.37\% | 43.46$\pm$0.24\% | 44.96$\pm$0.32\% |

---

### Official Review · Reviewer_92hs · 2021-07-16

**Rating:** 7
**Confidence:** 4

**Summary:**

Adversarial robustness has received a lot of attention along with the study of adversarial data. So far, almost works have shown that robust classifier not only outperform under comprehensive adversarial attack evaluations but also boost the performance in some downstream tasks. However, the underlying mechanism of adversarial robustness is still not clear.

In this paper, they investigate adversarial robustness from the perspective of linear components. From this very novel perspective, they find that there are some statistical properties for comprehensively robust classifiers. Specifically, robust classifiers show an obvious hierarchical clustering effect on their linear sub-networks. Based on the above investigation, this paper applies this on more tasks, such as robustness boosting and domain adaption.

**Ethical Concerns:**

No.

**Limitations And Societal Impact:**

The authors have discussed limitations and potential social impacts well.

**Main Review:**

Pros:

1. There are rare works to deeply think about the underlying mechanism of adversarial robustness. In this paper, they consider understanding this from the perspective of linear components. From this very novel perspective, they find that there are some statistical properties for comprehensively robust classifiers. Specifically, robust classifiers show an obvious hierarchical clustering effect on their linear sub-networks. This contribution is novel and has a significant impact on the field of adversarial machine learning.

2. They present an intriguing phenomenon of hierarchical clustering effect in robust models and provide a novel yet insightful understanding of adversarial robustness. The clustering effect aligned with class hierarchy demonstrates more semantic and representative feature extraction capacity of robust models, which benefits a lot in various tasks.

3. The paper is technically sound and well written. The proposed solution is based on a very interesting investigation. Figures are very informative.

4. Besides adversarial-related study, this paper further explores some downstream tasks with the understanding of hierarchical clustering. Experimental results show that the clustering effect and hierarchical classification learned by robust model benefits the task as well.

Cons:

1. The major concerns include the setup of experiments. The domain adaptation is a well-developed area and problem. This paper said their contributions to the domain adaptation field but did not explain details in such experiments. In domain adaptation, we focus the performance on the target domain. It is very confusing what kind of domain adaptation tasks is considered in Table 2. Why is there accuracy on the source domain? This is not the one cared about in the field.

2. I noticed many hyper parameters used in the experiments like dynamically changed learning rate. What are the motivations to design such ways to adjust hyper parameters?

3. It is unclear why the normalization in Eq. (8) is necessary. More explanations should be given.

4. In the defense part (Table 1), which method does this AT mean? It would be better to see the effectiveness of “+C” with more adversarial training methods.

--------------------------------------------
My concerns are addressed well. Thus, I will increase my score by 1 point.

**Time Spent Reviewing:**

7hrs

---

> ### Author Response · Authors · 2021-08-10
> **Response to Reviewer 2**
>
> Thanks for your detailed reading and suggestions.
>
> **Q1.** The confusing setting of domain adaptation tasks.
>
> **A1.** Thanks for your suggestions. In fact, in our setting, we divide the data into training and test ones following lines \#274-\#275. Then the source training data and target test data have different fine labels but share the same coarse labels. We add the accuracies in source domain to show more backbone differences for non-robust(NR) and robust(R) models.
>
> **Q2.** Why dynamically adjust hyperparameters such as learning rate?
>
> **A2.** The learning rates are chosen for training better models. In fact, we do not dynamically change learning rates, which are different when training from scratch (lr:0.1) or fine-tuning (lr:0.01). AlexNet is a light-weighted network thus we use a smaller learning rate of 0.01.
>
> **Q3.** Why do we need to normalize the weight vectors in Eq. 8?
>
> **A3.** In Sec. 3.1, we compute a correlation matrix $C$, which in fact explores the cosine similarity of extracted weight vectors. To compute the cosine values of weight vectors and avoid the class-wise scale variance, we first need to normalize all of them. This is also explained in lines \#140-\#141.
>
> **Q4.** What does AT in Table 1 refer to? Can '+C' operation be used in other adversarial training methods?
>
> **A4.** AT represents for Madry's adversarial training [23]. Yes, we add more results on TRADES here and our method brings robustness improvement. To note that, although our method can be generally applied with adversarial training methods and can also bring some improvement, in this paper we focus more on the understanding study as well as the further benefits brought by robust models ($\textit{e.g.}$, downstream domain adaption tasks).
>
> | ResNet-18s (CIFAR-10) | TRADES  | TRADES+C(Ours) |
> | --------------------- | ------- | -------------- |
> | ACC against PGD-20    | 52.80\% | 53.71\%        |

---

> > ### Comment · Reviewer_92hs · 2021-08-16
> > **Thanks for addressing my concerns**
> >
> > My concerns are addressed well. I will increase my score by 1 point.

---

> > > ### Author Response · Authors · 2021-08-18
> > > **Thanks**
> > >
> > > Thanks a lot for your valuable time and effort. The mentioned concerns are necessary for this paper. We will carefully address them and adopt the detailed suggestions in the revision.

---

### Official Review · Reviewer_cKPT · 2021-07-16

**Rating:** 6
**Confidence:** 3

**Summary:**

This paper introduces the "clustering effect" of linear components of adversarial robust models. To be specific, the authors propose to remove the non-linear functions in networks and further integrate all linear components into a single weight matrix W in order to directly study the effect of linearity in adversarial models. By conducting experiments with W, the authors find that adversarially trained models tend to show clustering effects in both weight and feature dimensions. This effect further benefits some downstream tasks such as adversarial attack and transfer learning and can be enhanced with additional supervision. In all, the phenomenon showed in this paper is interesting and deserves more and deeper research.


**Limitations And Societal Impact:**

The paper does not discuss the limitations. I think the most critical weakness is the too simple setting of the experiments and lack of deeper analysis. Please also see the main review for other weaknesses.


**Main Review:**

  - Originality: It seems novel -- I don't see the proposed method before.
  - Quality: Though the phenomenon shown in the paper is interesting, first of all, the setting is relatively simple, especially considering that the superclasses in most experiments are only animals and non-animals. Also, I do not see a convincing reason for ignoring fine-level accuracy in transfer learning from the authors. It would be good to explain why just compute the coarse-level accuracy. Second, the authors only point out this phenomenon and conduct further experiments on its applications without deep analysis on the reason behind it. I would expect more theoretical analysis on why linear components of the networks can show such clustering effects when trained with adversarial examples. Third, the experimental results are not always consistent and sometimes the improvement is also trivial which further enlarge my concern about the paper. For example, in Figure 5(a), we can see those non-robust models perform the worst as we would expect in certain situations but perform the best at other times. Also in Table 2, the performance after fine-tuning might be affected by the upper bound the backbone can achieve which can not fully support the idea that robust models can benefit more from fine-tuning.
  - Clarity: I have two questions that need to be further answered. First of all, how the proposed regularization loss is added to the training pipeline should be stated more clearly. According to Algorithm 2, after computing the regularization loss before retraining the model, is the feature clusters and the loss fixed or it is computed by the running mean of the samples? Besides, why the performance of AlexNet with R+C in Table 2 is so much higher than NR and R? (91 v.s. 85). I am wondering if this is not an expected phenomenon so some models might not be trained well enough. The writing of the paper needs to be improved greatly. Some paragraphs are hard to follow such as Section 4.1.1 and 4.2.1. There also exist lots of typos that need to be fixed. In the end, I would like to suggest the authors re-draw Figures 5 and 6. It is a little bit hard to distinguish all the shapes and colors without annotations on the graph when the only guidance is the text in the main body.
  - Significance: I think the phenomenon observed in this paper is quite interesting and deserves more research. It might be a good start point for this area if the paper can be updated.

**Time Spent Reviewing:**

5

---

> ### Author Response · Authors · 2021-08-10
> **Response to Reviewer1**
>
> Thanks for your valuable suggestions.
>
> **Q1.** Why not compute the fine-level accuracy of the target domain in domain adaptation tasks?
>
> **A1.** In our domain adaption setting, according to line \#239, the target data and source data have different fine labels (but share the same coarse labels). Therefore, we can not compute the fine-level accuracy directly in target domain. Such a division fashion is clearly stated in lines \#274-\#275. Taking Table 2 for an example, the training source data is composed of 4\*4 fine classes (The first 4 represents randomly selecting four coarse classes, the second 4 represents four fine classes randomly sampled from each coarse class) and the test target data is composed of 4\*1 fine classes (4 represents the selected four coarse classes, and 1 represents the left one fine class out of each coarse class). To be more specific, in Table 2, the left 4\*1 test target data have fine label index (91: Trout, 92: Tulip, 87: TV set, 98: Woman).
>
> **Q2.** Theoretical analysis on clustering phenomenon in adversarial training.
>
> **A2.** Intuitively, adversarial training helps model to extract more robust features, which tend to be more aligned with the shape information instead of texture one. Such features could be more semantic and tend to show a stronger connection with data hierarchy. Therefore the hierarchical clustering effect occurs in adversarial training. As for the theoretical analysis, we will refer to more studies on the connection from semantics to adversarial robustness by analyzing classifiers that employ clustering in representation space. And we will try to finish this in our future work.
>
> **Q3.** In Figure 5(a), the performance of non-robust models is sometimes good and sometimes bad.
>
> **A3.** In Figure 5(a), the $\texttt{circles}$ represent for non-robust models in the left panel. In the right panel, the $\texttt{pentagons}$ represent for non-robust models after fine-tuning. We will change the color and shape in Figure 5 for better vision. The performance of non-robust models is not sometimes good and sometimes bad. They always perform worst in target domain.
>
> **Q4.** The performance of robust models with fine-tuning is affected by the upper bound the backbone can achieve.
>
> **A4.** Thanks for your detailed reading. In Sec. 4.2.1 and Table 2, we try to show the overall superiority of robust(R) models compared to the non-robust(NR) ones in domain adaption tasks, which keeps consistent before or after fine-tuning. Take robust(R) ResNet-18 in Table 2 for example, the source coarse accuracy drops from 87.94\% to target coarse accuracy 82.50\%, and finally comes to 94.40\% after fine-tuning. Yes, the fine-tuning effect is affected by the robust backbone in a certain degree. But the comparison of model performance we indicated should start from the source coarse accuracy.
>
> **Q5.** How to obtain the regularization loss in Algorithm 2?
>
> **A5.** Sorry for the confusion. The regularization loss function is computed by Eq. 11 along with model training, where the class hierarchy information is achieved by $C_\mathrm{op}$ during some pre-training process.
>
> **Q6.** Are AlexNets shown in Table 2 not trained well enough?
>
> **A6.** As we described in Sec. 4.2.1, the training strategies used by different models are the same, even for AlexNet. And we used the best checkpoints. In addition, we also train an AlexNet model on CIFAR-20, which has the fine accuracy of 55.95\%. The similar accuracies indicate that the AlexNet models in Table 2 should be trained well enough.
>
> **Q7.** The writing of the paper needs to be improved. Figures 5 and 6 need to be drawn more clearly.
>
> **A7.** Thanks for your suggestions. We will further improve the writing and re-design Figures 5 and 6, especially with proper shapes and colors in the revision.

---

> > ### Comment · Reviewer_cKPT · 2021-09-01
> > **Thanks for the response!**
> >
> > Most of my concerns have been addressed. I would like to increase my score by one point (from 5 to 6).

---

### Decision · Program_Chairs · 2021-09-28

**Decision:**

Accept (Spotlight)

**Comment:**

This paper presents an interesting study of adversarial models that seems novel and creative.
The phenomenon identified for the linear components is intriguing, though it seems it remains speculative to fully understand its implications on the original network. Also, the experimental results could be stronger.
Nevertheless, given the novelty of proposal in an area that has received significant attention, this seems like an interesting contribution.

**Consistency Experiment:**

NeurIPS has a long history of experimentation. In 2014, NeurIPS ran an experiment in which 10% of submissions were reviewed by two independent committees to quantify the randomness in the review process. This year, we repeated a variant of this experiment to see how the quality of the review process has changed over time.  This paper was part of the experiment and was therefore assigned to two committees (consisting of reviewers, an Area Chair, and a Senior Area Chair) that reached independent decisions.  If both committees made the same recommendation, this recommendation was followed. If a single committee recommended acceptance, the paper was accepted (with the exception of a few cases in which the other committee identified what we considered a fatal flaw, e.g., an error in a key result).

This copy’s committee reached the following decision: **Accept (Poster)**

The other committee assigned to the paper recommended **Accept (Spotlight)**.  You can find the other set of reviews, along with any follow up discussion with the authors here:
https://openreview.net/forum?id=Xhj3PdCf4q9